# Do religious beliefs influence concerns for animal welfare? the role of religious orientation and ethical ideologies in attitudes toward animal protection amongst Muslim teachers and school staff in East Java, Indonesia

Dexon Pasaribu[1]☯*, Pim Martens[1,2‡], Bagus Takwin[3‡]

1 Maastricht Sustainability Institute, Maastricht University, Maastricht, The Netherlands, 2 Ethics of the Anthropocene Program, Vrije Universiteit, Amsterdam, The Netherlands, 3 Department of Social Psychology, Universitas Indonesia, Depok, Indonesia

☯ These authors contributed equally to this work.
‡ PM and BT also contributed equally to this work.
* dexon.pasaribu@maastrichtuniversity.nl, dexon.pasaribu@gmail.com

**Data Availability Statement:** All relevant data are within the paper and its Supporting Information files.

## Abstract

There is ample research supporting White's (1967) thesis, which postulates that religion and religious belief inhibit ecological concerns. This study thus seeks to explore the relationship between individuals' acceptability for harming animals as one representation of ecological concern (measured using Animal Issue scale (AIS)) and their religious belief (measured using Religious Orientation Scale (ROS)) and ethical ideology (measured using Ethical Position Questionnaire (EPQ)). The study surveyed 929 Muslim teachers and school staff in East Java, Indonesia. We found that ROS correlates with EPQ whereby intrinsic personal (IP) relates with idealism while extrinsic social religious orientation (ES)—where religion is perceived as an instrument for social gain, membership and support—relates with relativism. However, using multiple regression analysis to examine both EPQ and ROS relation to acceptability for harming animals suggests mixed results. We found that, idealism and IP relate to a lower acceptability for harming animals, while relativism and ES correlate to a higher acceptability for harming animals. In another model where we calculate all the main variables with all the demographical and other determinants, we found that only ROS consistently relates to acceptability for harming animals. Additionally, we identify, explain and discuss significant demographic determinants along with this study's limitations.

## 1. Introduction

Animals—specifically, beliefs and attitudes towards them—have a central role within the field of human-animal relationships, animal welfare, ecological belief and sustainability. Most

**Funding:** The work by DP has been made possible by Government of Indonesia Ministry of Finance's Endowment Fund for Education Institution's (LPDP) (Grant no 2017022221010339) scholarship and funding, https://www.lpdp.kemenkeu.go.id. The funders had no role in study design, data collection and analysis, decision to publish, or preparation of the manuscript.

**Competing interests:** The authors have declared that no competing interests exist.

research about belief and attitude shows that attitudes toward animals are closely related to some determinants such as age [1–3], household income [4], education [5–7], pet ownership [8–10], geographic region [11, 12] and religion [8, 13–15]. Regarding the latter, despite limited studies on the relationship between religious belief and public attitudes toward animals, there are growing investigations which confirm the relationship between ethical ideologies and public attitudes towards animals [16–20] as well as between ethical ideologies and religious orientation [21–24].

For the relation between ethical ideologies and public attitudes toward animals, previous studies showed mixed results. It was found that ethical idealism relates positively to a higher concern for animal use [16]. Through their research about the effectiveness of materials designed to sway public opinion about biomedical research using animals, Herzog & Nickell [17] would later add that compared to males and those low in ethical idealism, females and subjects high in moral idealism rate higher effectiveness to those research materials and advertising that reject animal use in biomedical research (anti-animal research materials) (p. 9). Also similar, Wuensch and Poteat [25] concluded that support for animal research associate negatively with idealism but positively with relativism. More recent studies by Su & Martens [18, 26] also confirmed these results, showing that higher idealism scorers are more likely to have a more positive attitude toward animals and a lower acceptability for harming animals. However Su & Martens [18, 26] slightly deviate from older studies [16, 17] whereby they find that high scorers of ethical relativism are more likely to have a more negative attitude toward animals only in China [18], but not in their Dutch sample [20]. Su & Martens argued that the differences between both samples might stem from the difference between being a developed and developing country, respectivily [18]. On the opposite spectrum, Wuensch, Jenkins, & Poteat [19] found that among nonidealists there is a significant positive relationship between misanthropy and support for animal rights, while among idealists the regression line is flat. They argued that misanthropic nonidealists discount the value of benefits to humankind (or may even consider them of negative value), and thus cannot justify animal use to benefit humankind [19].

For the relation between ethical ideologies and religion, previous studies' results were much more consistent and straightforward. Cornwell et al. [23] found that religion has some effect on ethical positions. Austrian Christians are significantly less idealistic and relativistic than all other religions, even with other Christians from the United States and Britain [23]. They argued that there are some ethical convergence between religions [23]. In another study, Barnett, Bass, and Brown [21] concluded that religiosity correlates positively with a non-relativist ethical ideology. Closely similar with them, Watson, Morris, Hood, Milliron, & Stutz [24] argued that religious intrinsicness or religious intrinsic personal orientation is associated with the idealism and antirelativism of an absolutist ethical position. They argued that intrinsic commitments to religion may simply mean that certain beliefs are absolutely nonnegotiable [24] (p. 5). In Forsyth's [27] terms, this absolutistic way of thinking type is the result when people strongly believe that moral decision should be guided by an universal governing principle (low relativism) rather than by personal or situational analysis (high relativism) while also convinced that ethical behavior will always lead to positive consequences. Thus, combining results from these studies, the present study aims to examine the relationship between ethical ideologies and religious orientation, and to explore how both religious orientation and ethical ideology interact and relate with participants' acceptability towards harming animals.

The role religion plays in studies of animal welfare remains unclear in some areas. One area, focusing on the correlation between religious affiliation and the treatment of animals, showed mix results. In some cases, religious practice is negatively correlated with positive animal attitudes [20, 28, 29], whilst in others there are no significant differences [17, 30]. Another

area focuses on the kind of animal being put into consideration. Gilhus [14] stresses the relevance of the value of animals within different religions, which is supported by Driscoll [8] a decade before, who argued that the acceptability in various scenarios of animal use strongly depends on the species of animal used in each of the provided examples. Lastly, another area put emphasis on the liberal-conservative continuum of ideology. Bowd & Bowd [13] showed that religious affiliation consistently correlated with attitudes toward the treatment of animals, and the level of attitude adopted (whether more or less humane) was related to the theological position of the affiliated church [13]. Participants who affiliated with theologically liberal religious groups tend to display more positive attitudes toward animals compared to those who affiliated with more conservative religious groups [13]. However, Driscoll [8] argued that persons who reported either an affiliation with or no religious affiliation with the Catholic church, rated examples of animal use as significantly less acceptable than persons who proclaimed a traditional Protestant affiliation. This was partially supported in Díaz [15], providing evidence that non-Catholics and non-practitioners of any religion were associated with the most positive attitudes toward animals. Similarly, Su & Martens [20] find lesser positive attitudes toward animals in respondents whose main source of inspiration was Christianity, as compared to those respondents who did not report Christianity as their main source of inspiration.

Despite these efforts, religion has barely been featured amongst key anthropogenic factors causing environmental degradation [31]; at least not until after White's [32] thesis about religion gained sufficient attention from the scientific community, where much of the later research would then assume that religion and ecology are interrelated. Several studies show that more often than not, religion hinders the awareness of and efforts towards environmental sustainability, where it depresses concern about the environment [33–35]. Others, however, have found that the belief in God or the identification with a particular religion is not associated with measures of environmental concern [36–39]. There are several possible reasons for these mixed results. One reason might stem from how each study addresses different aspects and properties of religion in measuring religious value, such as religious scriptures, contents and interpretation [40–42], or communication framing [39, 43]. Another reason might reside in how various studies differ in how they define religiosity, religiousness or religious belief. Gallagher & Tierney [44] argue that religiosity and religiousness are interchangeable as far an individual's conviction, devotion and veneration towards a divinity is concerned. However, religiosity or religiousness can be broadly or narrowly formulated using differing aspects such as (1) human cognitive aspect (beliefs, knowledge), (2) affect, which relates emotions to religion, and (3) behavior, such as time spent praying or reading religious texts, attendance, or affiliation [45]. Thus, differing foci and aspects produced various operationalizations of religiosity, such as religious orthodoxy [46, 47], typology [48], fundamentalism [49, 50], and religious orientation [51–53].

The present study utilizes Allport's religious orientation in defining the interchangeably-used religiosity or religiousness, as far as it approaches beliefs, knowledge and affectation of intrinsic, extrinsic personal and extrinsic social motivation in engaging in religious activities. In detail, Allport's religious orientation consists of intrinsic religious orientation, where religion is deeply personal to the individual, such as the commitment to a religious life and living out his/her religion; extrinsic personal religious orientation, with religion being a source of peace safety and comfort, which is a direct result of participating in religious activity; and, finally, extrinsic social religious orientation, where the emphasis is placed on religion as membership in a powerful in-group, providing protection, consolation or social status, and enabling religious participation [52, 54–57].

In other areas, studies examining the relationship between religious belief and ethical ideologies [22, 24, 58] provide evidence that ethical ideologies facilitate broader philosophical

coverage corresponding to religious values and beliefs [58]. Several studies argue that general spiritual principles and values are largely related to ethics [58–60], indicating that religiosity significantly correlated with Forsyth's [27] idealist and anti-relativist ethical ideologies [21, 24].

Forsyth's [27] ethical ideologies consists of two components, namely, ethical idealism and ethical relativism. An idealist thinks that ethical behavior will always lead to positive consequences, while a relativist rejects universal moral principles, instead believing that moral decisions should be based on a personal or situational analysis [27]. Several studies of ethical ideologies and attitudes towards animals and animal protection demonstrate that public attitudes toward animals or animal experiments are related to their ethical perspectives. One study investigating the role of idealism and relativism in the United States demonstrates how idealists often express greater moral concern for how animals are utilized than their relativist counterparts [25]. Later studies provide more evidence that positive attitudes towards animals are positively correlated to ethical idealism, where people's moral idealism significantly influences their attitudes toward animals [16, 18]. The more those individuals consider their ethical behavior would always lead to desirable consequences, the more they appreciate animals [18].

Nonetheless, the role religion plays regarding attitudes towards animals is as yet still unclear. Most studies of ethical ideologies provide reliable evidence that the position of ethical idealism bears positive attitudes towards animals and animal protection [16, 18, 20, 25]. Moreover, research on ethical ideologies also provide clear evidence where religiosity significantly correlates with idealism and anti-relativism [21, 24]. Thus, the present study aims to utilize ethical ideologies to examine the relationship between religiousness—as a major driver of ethics [58]—and public attitudes towards animals. This study aims to also take into account, therefore, how both religious belief and ethical ideology interacts with attitudes towards animals and their welfare and protection.

As it was found in previous studies [21, 24], as the first working hypothesis, we predict that intrinsic personal religious orientations will have a positive correlation with ethical idealism and a negative correlation with relativism. Also taking the consistent results from various studies [16, 18, 20, 25], as our second working hypothesis we predict that higher acceptability for harming animals relates to a lower ethical idealism and a higher relativism. The third working hypothesis is the extension of the first hypothesis, in which it predicts how religious orientation relates to attitudes toward animals and animal protection by examining how it correlates to ethical ideologies. We hypothesize that Allport's intrinsic personal religious orientations will have positive correlations to lower acceptability for harming animals.

In later developments of religious orientation [53, 57, 61], the dimension of extrinsic social motives has been added. Extrinsic social religious orientation addresses how individuals practice religion more as an instrument for social gain such as membership in a powerful in-group, providing protection, consolation or social status, and enabling religious participation. The extrinsic social religious orientation is more closely related to the social identity in-group membership concept [62–64] which introduce instrumental views of religion for social gain whereby religious belief systems are used to obtain desirable outcomes that might unnecessarily be ethical or unethical. On one hand, the ethical means for social gain may very much corresponds to the concept of ethical idealism where ethical behavior is believed will always bring positive outcome. However, on the other hand, should there be unethical means for social gains, it may relate to lower idealism, and higher relativism in which a person strongly believe that there is no universal moral standard, and therefore, moral decisions should be based on the personal or situational analysis. In this sense, we are carefully posing a working hypothesis for the relationship between extrinsic social religious orientation and ethical ideologies. Thus, as the fourth hypothesis, we predict that higher extrinsic social religious orientation relates to a

lower idealism and higher relativism position, whereas higher relativism relates to a higher acceptability for harming animals.

The observation that extrinsic social religious orientation overlaps with the social identity in-group membership concept [62–64] shows how important the concept of social category is. In this study, the religious group is treated as a social category that offers a sense of group positioning within which individuals identify themselves vis-à-vis religious outgroups [65, 66]. Thus, individuals who identify themselves as Muslims are more likely to behave in accordance with the typical behaviors of fellow Muslims. Thus, applying the above findings to the context of Indonesia, the present study avoids describing Islamic religious worldview of animals. Despite it being true that the majority of people in Indonesia follow Islam, this investigation is not theological in nature. Moreover, it is important to mention that this study purposefully selects the population in East Java province, depicting considering that it represents some of the oldest, most influential Islamic communities and organizations, whilst also being the province with the most diverse Islamic denomination. The province of East Java is the birthplace of Nahdlatul Ulama (NU), the largest Islamic mass organization in Indonesia. It has approximately 40 million members throughout the nation and its influence is not merely at the regency-level but also at the national [67]. Secondly, East Java is well-known for its long history of Islamic boarding schools. Pesantren Darul Ulum is one of the oldest and most distinguished in Jombang, East Java [68]. Thirdly, East Java offers an interesting segment of the political constellation in Indonesia. Its political influence at the national level has been prominent since the making of the nation [69]. Two of the most renowned instances were the appointment of Abdurrahman Wahid as the fourth President of Indonesia (1999–2001) and the appointment of Ma'ruf Amin as the current Indonesian vice president (took office in 2019), both of whom have strong ties to Nahdlatul Ulama in East Java. All in all, the above reasons foster East Java as one of the most relevant candidate-grounds for scrutinizing the relationship between religiousness and the attitudes held towards animals and animal protection; moreover, due to the religious groups' prevalence in East Java, we should point out that our respondents are likely to be Muslims. Regardless of all the above, however close a representation East Java is of the everyday major religious worldview in Indonesia, the present study avoids over-generalization of the results representing the whole country.

Aside from the above hypotheses, we also emphasize the demographic determinants commonly suggested in most studies about religion, ethical ideologies and animal welfare, such as gender, age, household income, education, pet ownership, religious organization affiliation, meat consumption [18, 20]. We will therefore closely scrutinize these important demographic or other determinants in our analysis.

## 2. Materials and methods

We confirm that this article was reviewed and approved by the institutional review board (ethics committee). We have submitted the plan for conducting the study, the time schedule, the questionnaires and the tools for collecting data and acquired the approval from the Maastricht University's Ethics Review Committee Inner City faculties. This research article conforms ethics for human participant regulated by the General Rules for Information Protection (European Union) 2016/679. All personal information is handled with extreme care so that personal data will not be opened to third parties or stored on servers that are accessible to public. Names and position is replaced by an alphanumeric code to keep identity protected.

We wrote an invitation letter to each school requesting their willingness to participate. This invitation letter was formalized and legalized by the relevant body of Indonesia government ranging from national, province to districts. All schools/universities that rejected our

invitation were not surveyed. For each of those schools/universities that accepted and were surveyed, we re-confirmed each participants' willingness to participate by obtaining the oral consent that they are freely and voluntarily participating in the survey.

This research targeted Muslim teachers and school staff in the province of East Java, Indonesia, using cluster sampling, whereby a paper and pencil survey of teachers was conducted. One of the reasons for the participant selection is in viewing that as an institution, both public and private schools are subjects to nation-wide education curriculum whereby collected data may generally capture a nation-wide curriculum's learning goals [70] relevant to animal protection and welfare. However, there were also a lengthy discussions about educators roles as transformative intellectuals rather than as nation-state agent teaching nation-state learning goals [71–73]. Also, taking some roles and responsibilities of a parent (*loco parentis*), teacher may be as well provide assistance and insight on moral, political, religious and ethical issues for their students [74] as one study hinted that teachers act as role-models for the students and influence their students' political attitudes [75].

In another study related to transformative agency, teachers' inclusive practices, moral purposes, competence, autonomy and reflexivity [76] are important factors to act as an agent of change. The duality of being transformational agents while also fulfilling their obligatory role to implement the nation-state education curriculum agenda, Muff & Bekerman [71] argued that teachers mediated their roles between the different demands that of the civic education politics imposes on them by navigating elegantly both in producing hegemonic discourse and in fostering ways to rebel against and draw counter-hegemonic strategies in their classroom practice. Thus, this study viewed that having teachers as the participants for the research would capture some dynamics of interlocking roles at play. To name a few, the nation-state curriculum goals, teachers' beliefs, moral purposes, reflexivity and awareness in responding to the nation-state curriculum, and their combined roles as transformative intellectuals, more or less, are the dynamics reflected in classroom discourses. Teachers attitudes towards animal welfare and protection may best represent the nation's sets of environmental policy and the younger generation's perspective.

Survey participation invitations were sent to 67 schools (ranging from junior to senior high schools). The survey invitation emphasized that it was important for the school to provide a balanced proportion of male and female teachers or school staff. Total of 37 schools, from 10 districts of East Java, replied and agreed to participate, providing 1007 participants. However, only 929 participants were analysed due to removing 78 participants because of incomplete and unengaged answers (*see section 3.2*).

All the questionnaires in the survey were originally in English (*see* S1 Appendix). We then translated them to Indonesian (*see* S2 Appendix). The method of translation and adaptation was using expert judgement and back translation. The questionnaires were translated to Bahasa Indonesia and sent to experts for evaluation and finalization of the translation. After corrections, the questionnaires were translated back to English by three Indonesian academicians from Universitas Indonesia. Back-translated items that were very similar to their English language origin were retained, and the remaining were modified or deleted.

The set of questionnaires consist of four sections. In the first section, we asked a variety of important determinants and demographic details such as birth year (age), gender, highest level of education completed, their experience or participation in either animal protection, nature conservation, or human health organization, their household composition (for example, single, married, or widow(er), with children or not), place of residence (rural or urban), type of house (apartment, live with parents, etc.), their opinion regarding the importance of religion/ spirituality in their lives, household income, pet ownership, kinds of pet, their weekly frequency of meat consumption, and the frequency of visiting public zoos or aquariums in a year.

In the second section, the Animal Issue Scale (AIS) [30] is used to measure acceptability toward harming animals. There are 43 questions in the original AIS, representing eight animal issues: use of animals, animal integrity destruction, killing animals, animal welfare deprivation, experimentation on animals, changes in animals' genotypes, harm animals for environmental reasons, and societal attitudes toward animals [harm animals for social issues]. Each question is rated on a five-point scale ranging from one, extremely unacceptable, to five, extremely acceptable. A high score on a question indicates a high level of acceptability for the particular issue [11]. Using principal axis factoring factor analysis (Tables 3–5 in S1 Data), the original 'killing animal' and 'animal deprivation' issues were identified as one factor (Table 1).

Thus, the present study reduced AIS to only 31 items, conveyed only 7 factors. Additionally, this study included the Animal Attitudes Scale (AAS) [77] for measuring public attitudes toward animals. However, after principal axis factoring factor analysis, previously intended as cross-validation for the AIS, the 20-item Likert-like scale AAS failed to provide a stable unidimensional construct as it was in its original psychometric properties (*see* Tables 35–49 in S1 Data). Alpha's reliability also showed small to moderate coefficients for each of the resulting factors. Thus, AAS was removed from the analysis.

In the third section, the Religious Orientation Scale (ROS) [51, 52, 78] was originally used to measure intrinsic and extrinsic religious orientation. We use Maltby's [57] 15-item version which incorporates Kirkpatrick's [79] analysis expanding ROS into three scales: intrinsic orientation (IP), extrinsic personal—religion as a source of comfort (EP) and extrinsic social—religion as social gain (ES). The 15-item scale therefore consists of nine questions addressing IP, for example, 'I try hard to live all my life according to my religious beliefs', 'My whole approach to life is based on my religion', 'It is important to me to spend time in private thought and prayer'); three questions addressing EP, for example 'Prayer is for peace and happiness', 'I pray mainly to gain relief and protection'; and lastly, the remaining three covering the ES dimension, for example, 'I go to church because it helps me make friends', 'I go to church mainly because I enjoy seeing people I know there'. However, after principal axis factoring factor analysis (Tables 21–26 in S1 Data), the present study found only two dimensions of intrinsic personal (IP) and extrinsic social (ES). After factor analysis, the EP was accounted as the same factor as IP (Table 2), and thus, will be considered as the same as IP.

In the fourth section, the Ethical Position Questionnaire (EPQ) was used to measure the differences in personal moral philosophy [16, 27]. The original EPQ was a 20-items Likert scale consist of two sub-scales. The first 10 items were designed to measure the ethical idealism dimension, while the last 10 items measured ethical relativism. Respondents were asked to respond to statement using the nine-point EPQ ranging from one (completely disagree) to nine (completely agree). Regarding the ethical idealism, six items were removed from analysis of this study. Four out of those six items were removed because of significant skew values which were outside the range between -2 to 2 [80]. The remaining two were removed because of low factor loading, along with three items from ethical relativism. After principal axis factoring factor analysis (Tables 27–34 in S1 Data), the present study uses only 11 EPQ items. In which four items from the idealism scale, and seven items from the relativism scale. Factor analysis also found that the remaining seven items of ethical relativism were put into two factors. However, after ensuring a relatively stable Cronbach alpha's reliability in one factor model, the present study decided to retain ethical relativism as it was, a one factor construct (model two, *see* Table 3).

## 2.1. Statistical analysis

Religious orientation, ethical ideologies and acceptability toward harming animals were analyzed with IBM SPSS 24 using multiple regression statistical procedures. This study also used

**Table 1. AIS rotated factor matrix.**

| Items | Factor[a] | | | | | | |
|---|---|---|---|---|---|---|---|
| | 1 | 2 | 3 | 4 | 5 | 6 | 7 |
| AI01_AnimUse Keeping animals for the production of food or clothing | | | | | | | .490 |
| AI02_AnimUse Keeping animals as pets | | | | | | | .447 |
| AI04_AnimUse Using animals for work | | | | | | | .624 |
| AI05_AnimUse Using animals for entertainment or sports | | | | | | | .654 |
| AI08_Intgrty De-sexing by hormone implants | | | | .542 | | | |
| AI09_Intgrty Removal of a body part, such as tail docking or de-clawing | | | | .662 | | | |
| AI10_Intgrty Marking animals by branding or ear notching | | | | .589 | | | |
| AI11_Intgrty Removal of dead tissue, such as hair/wool removal or foot trimming | | | | .557 | | | |
| AI14_Kill Using animals for products after their natural death | .439 | | | | | | |
| AI16_Kill Euthanizing healthy and unwanted pets because of overpopulation | .556 | | | | | | |
| AI17_Welfare Depriving animals of their needs for food and water | .768 | | | | | | |
| AI18_Welfare Depriving animals of an appropriate environment to rest, including shelter | .765 | | | | | | |
| AI19_Welfare Inflicting pain, injury or disease on animals | .798 | | | | | | |
| AI20_Welfare Not providing sufficient space, proper facilities and company needed for animals | .701 | | | | | | |
| AI21_Welfare Subjecting animals to conditions and treatment which cause mental suffering | .501 | | | | | | |
| AI24_Xprmnt Medical experiments using animals to improve human health | | | | | .553 | | |
| AI25_Xprmnt Testing cosmetics or household products on animals | | | | | .636 | | |
| AI26_Xprmnt Operating on living animals for the benefits of human medicine research | | | | | .755 | | |
| AI27_Genchng Increasing animals' reproductive or productive capabilities by genetic changes, eg cows producing more milk | | | .633 | | | | |
| AI28_Genchng Increasing animals' health or disease resistance by genetic changes | | | .693 | | | | |
| AI29_Genchng Creating farm animals that are more profitable because they feel happy with little stimulation and have little desire to be active | | | .749 | | | | |
| AI30_Genchng Genetic selection of pet animals, such as dogs and cats, to increase their rarity, potential for showing or pedigree value | | | .600 | | | | |
| AI34_EnvIss Controlling wildlife populations by killing | | | | | | .542 | |
| AI35_EnvIss Controlling animal populations by sterilization | | | | | | .439 | |
| AI36_EnvIss Destroying the habitat of endangered animal species | | | | | | .596 | |
| AI37_EnvIss Destroying the habitat of non-endangered animal species to develop and promote urbanization or crops to feed humans | | | | | | .465 | |
| AI39_SocAtt Considering some animal species as sacred or good luck symbols or totems | | .606 | | | | | |
| AI40_SocAtt Considering some animal species as evil or bad luck | | .765 | | | | | |
| AI41_SocAtt Parents displaying cruel treatment of animals in front of their children | | .591 | | | | | |
| AI42_SocAtt Inflicting pain or injury on animals as part of cultural traditions | | .570 | | | | | |
| AI43_SocAtt Cloning animals for human benefit | | .435 | | | | | |

Extraction Method: Principal Axis Factoring.

Rotation Method: Varimax with Kaiser Normalization.

a. Rotation converged in 7 iterations.

Pearson correlation product moment in investigating the relation between religious orientation and ethical ideologies. The resulting correlation tables provides additional explanation for the multiple regression results.

Previous studies examining the relation between EPQ public attitude toward animal and animal protection were conducted using ANOVA design [18, 20], where EPQ was considered as categorical variables differentiated into four groups depending on the high and low of each ethical idealism and relativism score. These groups are, situationists (high idealism and high

**Table 2. ROS rotated factor matrix.**

| Items | Factor[a] | |
|---|---|---|
| | 1 | 2 |
| ROS01 (IP) I try hard to live all my life according to my religious beliefs | .673 | |
| ROS03 (IP) I have often had a strong sense of God's presence | .608 | |
| ROS04 (IP) My whole approach to life is based on my religion | .705 | |
| ROS05 (IP) Prayers I say when I'm alone are as important as those I say in church | .577 | |
| ROS06 (IP) I attend church once a week or more | .358 | |
| ROS07 (IP) My religion is important because it answers many questions about the meaning of life | .741 | |
| ROS08 (IP) I enjoy reading about my religion | .750 | |
| ROS09 (IP) It is important to me to spend time in private thought and prayer | .630 | |
| ROS10 (EP) What religion offers me most is comfort in times of trouble and sorrow | .665 | |
| ROS11 (EP) Prayer is for peace and happiness | .764 | |
| ROS12 (EP) I pray mainly to gain relief and protection | .622 | |
| ROS13 (ES) I go to church because it helps me make friends | | .833 |
| ROS14 (ES) I go to church mainly because I enjoy seeing people I know there | | .894 |
| ROS15 (ES) I go to church mostly to spend time with my friends | | .787 |

Extraction Method: Principal Axis Factoring.

Rotation Method: Varimax with Kaiser Normalization.

a. Rotation converged in 3 iterations.

relativism), subjectivists (low idealism and high relativism), absolutists (high idealism and low relativism) and exceptionists (low idealism and low relativism) (Fig 1). In this study however, we view that it is best to retain the interval properties from the total score of ethical idealism and relativism to provide richer and a more detailed data. Thus, multiple regression is our selected statistical procedure for the given data.

This study uses two models of multiple regression. The first model only investigates the main variables, while the second model takes all main variables with the demographic and other important determinants. For both of the regression models, this study avoids stepwise method in considering that stepwise estimates are not invariant to inconsequential linear transformation. [81]. Rather, we follow Whittingham, Stephens, Bradbury, and Freckleton's [82] suggestion to use a full model including all of the effects (enter method) for the second regression model, where it takes all multiple variables (main variables, demographic and other determinants) which mainly consist of either interval or categorical properties. As a side note, this study converts all categorical variables into dummy variables, in which we expand each category as a new variables scored with either one or zero.

As Pearson correlation procedure is vulnerable from skewed and kurtosis distribution, we made preliminary normal distribution check to avoid inflated correlation. Each item in the questionnaire were checked for normal distribution assumption (Table 2; in S1 Data). In regards to normal distribution assumption, Kim [80] stressed that the tendency of large samples producing inflated z in consideration to large samples will usually produce a very small standard error for both skewness and kurtosis. Therefore, using skewness and kurtosis reference values for N more than 300, the present study removed items with kurtosis value outside the range between -7 to 7, or skew value outside the range between -2 to 2 [80].

After analyzing each items in the questionnaires, this study removed four items from EPQ idealism, which were "People should make certain that their actions never intentionally harm another even to a small degree", "One should never psychologically or physically harm another person", "One should not perform an action which might in any way threaten the dignity and

**Table 3. EPQ pattern matrix.**

| Items | Model 1 (using eigen value > 1)[ab] | | | Model 2 (forced as 2 factor loadings)[c] | |
|---|---|---|---|---|---|
| | **1** | **2** | **3** | **1** | **2** |
| EPQ02 (I) Risks to another should never be tolerated, irrespective of how small the risks might be. | | 0.57 | | | 0.52 |
| EPQ03 (I) The existence of potential harm to others is always wrong, irrespective of the benefits to be gained. | | 0.68 | | | 0.627 |
| EPQ08 (I) The dignity and welfare of the people should be the most important concern in any society. | | 0.563 | | | 0.584 |
| EPQ10 (I) Moral behaviors are actions that closely match ideals of the most "perfect" action. | | 0.453 | | | 0.48 |
| EPQ13 (R) Moral standards should be seen as being individualistic; what one person considers to be moral may be judged to be immoral by another person. | 0.742 | | | 0.459 | |
| EPQ14 (R) Different types of morality cannot be compared as to "rightness." | 0.679 | | | 0.491 | |
| EPQ15 (R) Questions of what is ethical for everyone can never be resolved since what is moral or immoral is up to the individual. | 0.757 | | | 0.624 | |
| EPQ16 (R) Moral standards are simply personal rules that indicate how a person should behave, and are not to be applied in making judgments of others. | 0.508 | | | 0.534 | |
| EPQ18 (R) Rigidly codifying an ethical position that prevents certain types of actions could stand in the way of better human relations and adjustment. | | | | 0.528 | |
| EPQ19 (R) No rule concerning lying can be formulated; whether a lie is permissible or not permissible totally depends upon the situation. | | | 0.882 | 0.729 | |
| EPQ20 (R) Whether a lie is judged to be moral or immoral depends upon the circumstances surrounding the action. | | | 0.727 | 0.673 | |

Extraction Method: Principal Axis Factoring.

Rotation Method: Oblimin with Kaiser Normalization.

a. Rotation converged in 7 iterations.

b. Suppressing values less than 0.4.

welfare of another individual", and "If an action could harm an innocent other, then it should not be done". Table 4 shows that all scales from the collected data is safely within the normal distribution bound. Thus, no transformation for normalization is needed.

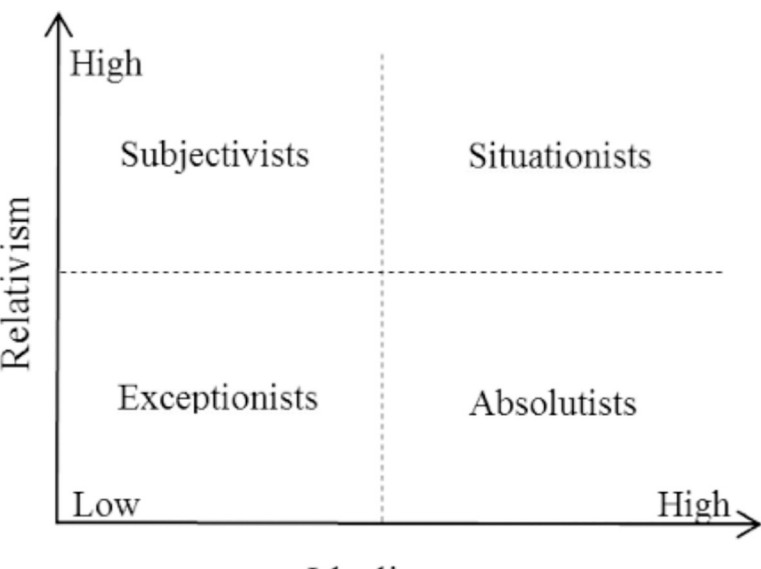

**Fig 1. Ethical positions according idealism and relativism.**

**Table 4. Skewness and kurtosis value of main variables.**

| | N | Skewness | | Kurtosis | |
|---|---|---|---|---|---|
| | Statistic | Statistic | Std. Error | Statistic | Std. Error |
| AIS | 929 | .389 | .080 | .939 | .160 |
| Animal use subscale | 929 | -.132 | .080 | .191 | .160 |
| Integrity destruction | 929 | .446 | .080 | .239 | .160 |
| Killing animal and animal welfare deprivation | 929 | .789 | .080 | .689 | .160 |
| Animal experimentation | 929 | -.250 | .080 | -.008 | .160 |
| Genotype change | 929 | -.426 | .080 | .463 | .160 |
| Harm animal for environmental issue | 929 | .418 | .080 | .022 | .160 |
| Societal attitude toward animal. | 929 | .565 | .080 | .289 | .160 |
| EPQ Idealism | 929 | -1.196 | .080 | 1.162 | .160 |
| EPQ Relativism | 929 | -.568 | .080 | -.017 | .160 |
| ROS Intrinsic Personal | 929 | -.751 | .080 | 1.430 | .160 |
| ROS_Extrinsic Social | 929 | .195 | .080 | -.495 | .160 |

Valid N (listwise) 929.

# 3. Results

## 3.1. Instruments validity

Table 5 provides the descriptive statistics for the variables used in the analysis. All the Cronbach's coefficient are acceptable, ranging from a moderate internal consistency value of 0.66 for the 'animal use' issue to a value of 0.91 for the overall animal issue scale.

The mean score for IP was 4.22 (SD = 0.53, with maximum score of five) indicating that, overall, the respondents considered themselves to be strongly committed to their personal religious life. The mean score for ES was 2.79 (SD = 0.99) indicating that, on the whole, the respondents were neither strongly nor weakly disposed towards viewing their religious practices as an instrument for social gain.

The mean idealism score of 7.2 (SD = 1.22, with a maximum score of 9) indicated that, in general, the sample had a strong idealistic ethical ideology, where they believe that their ethical

**Table 5. Descriptive statistics and measurement characteristics for variables.**

| Variable | Scale description | Number of items | Reliability | Mean | SD |
|---|---|---|---|---|---|
| ROS-Intrinsic Personal (IP) | 5-point Likert-like | 11 | 0.88 | 4.22 | 0.53 |
| ROS-Extrinsic social (ES) | 5-point Likert-like | 3 | 0.87 | 2.79 | 0.99 |
| EPQ Idealism | 9-point Likert-like | 4 | 0.66 | 7.2 | 1.22 |
| EPQ Relativism | 9-point Likert-like | 7 | 0.80 | 6.29 | 1.46 |
| Animal Issue Scale (AIS) | 5-point Likert-like | 31 | 0.91 | 2.54 | 0.52 |
| Animal use | 5-point Likert-like | 4 | 0.66 | 3.1 | 0.65 |
| Integrity destruction | 5-point Likert-like | 4 | 0.78 | 2.37 | 0.79 |
| Killing-welfare deprivation | 5-point Likert-like | 7 | 0.87 | 2.09 | 0.76 |
| Experiment | 5-point Likert-like | 3 | 0.82 | 3 | 0.83 |
| Genetic change | 5-point Likert-like | 4 | 0.8 | 3.3 | 0.75 |
| Harm for environmental issues | 5-point Likert-like | 4 | 0.75 | 2.37 | 0.79 |
| Harm for social issues | 5-point Likert-like | 5 | 0.84 | 2.11 | 0.76 |

*Using pearson correlation coefficient instead of Cronbach alpha, considering that the scale consists of only two items.

behaviour will always lead to positive consequences. The mean relativism score was 6.29 (SD = 1.46), indicating that on the whole, the respondents believe that moral decision-making should be based on situational, rather than universal principle.

The mean score of overall acceptability toward harming animal (AIS) was 2.54 (SD = 0.52), indicating that, in general, were neither strongly nor weakly disposed towards acceptability of harming animals. Except for the issues of animal use (mean of 3.1, SD = 0.65), experimentation (mean of 3, SD = 0.83) and genetic change (mean of 3.3, SD = 0.75), the remaining four dimensions of animal integrity destruction (mean of 2.37, SD = 0.79), killing-welfare deprivation of animal (mean of 2.09, SD = 0.76), harm (animals) for environmental issue (mean of 2.37, SD = 0.79), and harm (animals) for social issue (mean of 2.11, SD = 0.76) showed some tendencies to lean more to a lower acceptability of harming animals.

## 3.2. Response rates

From 1007 total responses obtained, 78 respondents (8%) were removed due to unengaged answers (in other words, these were the respondents who gave the same answer for all the questions in the questionnaire). After the removal, there were still some incomplete answers (listwise missing case) from for the remaining 929 participants (Table 1, in S1 Data). Those missing cases were imputed using a linear trend method. In total, this research collected and analysed 929 respondents. The mean age of all respondents (51% female (N = 475) and 49% male (N = 454)) is 36.38 years old (SD = 10.02). The completed surveys have a relatively balanced proportion of rural (61%) and urban (39%) areas. Additionally, several complementary variables were assessed, such as pet ownership, where 48% of respondents adopted one or more pet(s), while 52% of respondents didn't adopt any pet. For home ownership, 1% lived in apartment, 9% live in a rented room, 55% lived and owned a house, while the remaining 40% still live in their parent's house. For the highest level of education, 74% hold a Bachelor, 14% a PhD or a Master, 8% graduated high school, 3% hold a diploma, while for the categories of those who either finished middle or high school, where they either hold another degree, or did not answer, were each less than 1%. Regarding the frequency of zoo or aquarium visitation, 4% visited a zoo once a month, 7% at least every six months, 22% once a year, 42% once in every two or more years, and lastly, 22% never visited a zoo or aquarium, leaving the remaining 1% respondents without answer. Regarding professions, all of the respondents were teachers or school staff. However, some of the respondents had a secondary profession, as follows: 5% as an entrepreneur, 39% as an employee in the private sector, 24% as civil servants, 5% are also scholarship students, 19% are teachers or lecturers without a secondary profession, while the remaining 6% are either semi-retired, social workers, or university researchers, working in the farming or livestock sector; others did not disclose their professions, or did not or did not want to answer. Finally, we also asked about the frequency of weekly meat consumption whereby 6% didn't eat meat, 28% ate meat once in a week, 36% ate meat two to three days in a week, 13% four to six days in a week, and lastly, 14% ate meat every day.

## 3.3. Ethical ideologies and religious orientation

The hypothesis presented in this section is that higher personal religious orientation relates to a higher idealism and a lower relativism. Table 6 provides the correlation matrix for the studied variables. We find positive relationship between idealism with personal religious orientation (IP) (r[927] = 0.21, $p < 0.01$). However, there is no significant relationship between relativism with IP (r[927] = 0.000, $p > 0.05$), and therefore, while the hypothesis is rejected by every relation with relativism, it is accepted in predicting the relationship between idealism with IP. Lastly, the correlation between extrinsic social religious orientation and idealism (r

**Table 6. Correlation matrix between AIS, ROS and EPQ.**

| | AIS | | | IP | | | ES | | | EPQ Idealism | | |
| | r | CI 95% | | r | CI 95% | | r | CI 95% | | r | CI 95% | |
| | | lower | upper | | lower | upper | | lower | upper | | lower | upper |
|---|---|---|---|---|---|---|---|---|---|---|---|---|---|
| AIS | | | | | | | | | | | | | |
| IP | -0.19** | -0.25 | -0.12 | | | | | | | | | | |
| ES | 0.24** | 0.17 | 0.30 | 0.05 | -0.02 | 0.11 | | | | | | | |
| Idealism | -0.04 | -0.11 | 0.02 | 0.21** | 0.15 | 0.27 | -0.02 | -0.08 | 0.05 | | | | |
| Relativism | 0.15** | -0.21 | -0.08 | 0.00 | -0.06 | 0.06 | 0.15** | 0.08 | 0.21 | 0.35** | 0.29 | 0.41 |

**. Correlation is significant at the 0.01 level (2-tailed).

[927] = -0.02, $p > 0.05$) and relativism (r[927] = 0.15, $p < 0.01$) is reported with a more detail in another section (*see section 3.6*).

## 3.4. Ethical ideologies and acceptability for harming animals

The hypothesis presented in this section is that higher acceptability for harming animals (AIS) relates to a lower idealism and a higher relativism. There are two models developed and analysed using the multiple regression method (Table 7). The first model analyses the four main variables relation to AIS, namely idealism, relativism, intrinsic personal and extrinsic social religious orientation. The second model investigates all four main variables by taking all potential demographic and other determinants into account with the equation.

For the first model (Table 7), we find only partial evidence to support the hypothesis. The results show that only for ethical relativism we can accept the hypothesis, where we find higher relativism is more likely lead to a higher overall acceptability for harming animals (AIS) (b = 0.05, $p < 0.01$). This means that when holding all other variables constant, one point increase in relativism is likely to increase 0.05 point of AIS score. Moreover, it is important to mention that from the effect-size aspect, relativism has little to no effect toward AIS score ($F^2 < = 0.02$). Through the confidence interval, if we were to retake the regression for total of 20 random trials, taking samples of the same size from the same population, we can be confident that for 19 out of total 20 trials (95% of the time), an increase of 1 unit of relativism will be more likely to increase AIS between 0.02 to 0.07 point. Thus despite accepting the hypothesis for every relation with relativism, this study advises to take caution to limit the interpretation because of the near non-existent effect-size. In short, in the first model, idealism has no relation to overall acceptability for harming animals (AIS), and relativism significantly relates to a higher AIS (b = 0.05, $p < 0.01$). However, the confidence interval and effect-size indicate a small to no effect, suggesting that relativism relation to AIS is not as strong as its relation with IP, ES, and some of demographical or other determinants.

An important addition from this study is when observing the second regression model, whereby all main variables along with demographics and other determinants are taken together as well as independently. From the second model, this study shows no significant relation between AIS with both relativism and idealism.

## 3.5. Religious orientation and acceptability for harming animals

The hypothesis presented in this section is that higher intrinsic (IP) religious orientation relates to a lower acceptability for harming animals (AIS). In both of model (Table 7), the present study accepts the third hypothesis. We find that higher intrinsic personal religious

**Table 7. Regression of EPQ, ROS, and demographic determinants toward AIS.**

| Model | AIS b (Std. b) | | | Effect Size | | 95% CI Lower | 95% CI Upper |
|---|---|---|---|---|---|---|---|
| **1—Main Variable[A] (R = 0.33; R² = 0.11, df = 9,439)** | | | | | | | |
| (Constant) | 2.84 | | ** | | | 2.524 | 3.147 |
| EPQ Ideal | -0.02 | -0.04 | | 0.00[C] | | -0.05 | 0.01 |
| EPQ Relative | 0.05 | 0.13 | ** | 0.01[C] | | 0.02 | 0.07 |
| ROS Personal | -0.18 | -0.19 | ** | 0.03[C] | + | -0.25 | -0.12 |
| ROS Social | 0.12 | 0.22 | ** | 0.05[C] | + | 0.08 | 0.15 |
| **2—Main Variable + Demographic and other determinants[B] (R = 0.40; R² = 0.16, df = 40, 408)** | | | | | | | |
| (Constant) | 2.52 | | ** | | | 1.874 | 3.175 |
| [1]How often do you consume meat in a week? I don't consume meat: Yes (1)–No (0) | 0.36 | 0.18 | ** | 0.10[D] | + | -0.109 | 0.217 |
| [2]What is your gender? Female: Yes (1)–No (0) | -0.14 | -0.16 | ** | 0.22[D] | + | -0.18 | -0.05 |
| [3]What is the highest level of schooling you have completed? Diploma: Yes (1)–No (0) | 0.39 | 0.12 | * | 0.69[D] | + + | 0.16 | 0.61 |
| ROS Personal | -0.11 | -0.11 | * | 0.01[C] | | -0.200 | -0.014 |
| ROS Social | 0.05 | 0.11 | * | 0.01[C] | | 0.007 | 0.095 |
| [4]In what sort of house do you live? Own house: Yes (1)–No (0) | 0.11 | 0.12 | * | 0.16[D] | | 0.01 | 0.16 |
| What is your gross household expenses per month? Above 25 million: Yes (1)–No (0) | -0.50 | -0.07 | | - | | -1.130 | 0.131 |
| How often do you visit a zoo or aquarium? Once every six month: Yes (1)–No (0) | -0.15 | -0.08 | | - | | -0.343 | 0.045 |
| Where is your current residence place? Urban area: Yes (1)–No (0) | -0.07 | -0.08 | | - | | -0.169 | 0.025 |
| What is your gross household expenses per month? Refuse to answer: Yes (1)–No (0) | -0.08 | -0.07 | | - | | -0.193 | 0.034 |
| What is your age? | 0.00 | 0.08 | | - | | -0.002 | 0.010 |
| What is the highest level of schooling you have completed? Bachelor: Yes (1)–No (0) | 0.08 | 0.08 | | - | | -0.044 | 0.210 |
| What is your gross household expenses per month? Five to 10 million: Yes (1)–No (0) | -0.09 | -0.07 | | - | | -0.237 | 0.053 |
| Do you have children? Yes (1)–No (0) | 0.09 | 0.09 | | - | | -0.061 | 0.250 |
| In what sort of house do you live? Apartment: Yes (1)–No (0) | 0.25 | 0.06 | | - | | -0.167 | 0.665 |
| EPQ Ideal | -0.02 | -0.06 | | - | | -0.060 | 0.016 |
| EPQ Relative | 0.02 | 0.05 | | - | | -0.015 | 0.049 |
| How often do you consume meat in a week? Two to three times a week: Yes (1)–No (0) | 0.05 | 0.06 | | - | | -0.049 | 0.156 |
| What is your marriage status? Married: Yes (1)–No (0) | -0.09 | -0.08 | | - | | -0.267 | 0.088 |
| Is religion important for you? Yes (1)–No (0) | 0.19 | 0.05 | | - | | -0.189 | 0.572 |
| How often do you consume meat in a week? Once a week: Yes (1)–No (0) | 0.07 | 0.05 | | - | | -0.071 | 0.205 |
| Do you have pet? Yes (1)–No (0) | -0.04 | -0.05 | | - | | -0.134 | 0.046 |
| What is your marriage status? Widow(er): Yes (1)–No (0) | -0.16 | -0.05 | | - | | -0.523 | 0.199 |
| In what sort of house do you live? Room rent: Yes (1)–No (0) | 0.07 | 0.05 | | - | | -0.087 | 0.219 |
| Do you belong or donate to an organization or charity involved in or concerned with: Conservation of the natural environment: Yes (1)–No (0) | 0.08 | 0.05 | | - | | -0.115 | 0.267 |
| What is your gross household income per month? More than twice the average income in my country: Yes (1)–No (0) | 0.21 | 0.04 | | - | | -0.328 | 0.749 |
| What is your gross household expenses per month? 10 to 15 million: Yes (1)–No (0) | -0.20 | -0.04 | | - | | -0.718 | 0.319 |
| What is the highest level of schooling you have completed? Senior high: Yes (1)–No (0) | -0.07 | -0.04 | | - | | -0.269 | 0.128 |
| Do you belong or donate to an organization or charity involved in or concerned with: Improving health or human rights: Yes (1)–No (0) | 0.06 | 0.04 | | - | | -0.106 | 0.220 |
| How often do you visit a zoo or aquarium? Once every two or more year: Yes (1)–No (0) | -0.03 | -0.04 | | - | | -0.149 | 0.082 |
| Do you have your own backyard? Yes (1)–No (0) | -0.03 | -0.03 | | - | | -0.116 | 0.064 |
| How often do you consume meat in a week? Four to six times a week: Yes (1)–No (0) | 0.04 | 0.03 | | - | | -0.103 | 0.181 |
| What is your gross household income per month? Refuse to answer: Yes (1)–No (0) | 0.03 | 0.03 | | - | | -0.088 | 0.155 |
| What is your gross household income per month? About twice the average income in my country: Yes (1)–No (0) | 0.08 | 0.03 | | - | | -0.224 | 0.390 |
| Do you belong or donate to an organization or charity involved in or concerned with: Animal sector: Yes (1)–No (0) | -0.05 | -0.02 | | - | | -0.339 | 0.242 |
| How often do you visit a zoo or aquarium? Once a month: Yes (1)–No (0) | -0.04 | -0.02 | | - | | -0.288 | 0.211 |

*(Continued)*

**Table 7.** (Continued)

| Model | AIS b (Std. b) | | Effect Size | 95% CI | |
|---|---|---|---|---|---|
| | | | | Lower | Upper |
| How often do you visit a zoo or aquarium? Once a year: Yes (1)–No (0) | -0.02 | -0.02 | - | -0.148 | 0.112 |
| What is your gross household income per month? About the average income in my country: Yes (1)–No (0) | -0.01 | -0.01 | - | -0.138 | 0.115 |
| Do you have any affiliation to religious organization? Yes (1)–No (0) | -0.01 | -0.01 | - | -0.116 | 0.102 |
| What is your gross household income per month? About the minimum income in my country: Yes (1)–No (0) | 0.01 | 0.01 | - | -0.130 | 0.146 |

[*]$p < .05$

[**]$p < .01$

[A]regression using enter method in a stepwise manner

[B]regression using enter method

[C]effect-size calculation using eta squared ($F^2$)

[D]effect-size calculation using Hedge's g; +small effect size $F^2 > = 0.02$ (or in some cases of categorical dummy variable, using Cohen's D/Hedges'g $> = 0.2$); ++medium effect size $F^2 > = 0.15$ (or in some cases of categorical dummy variable, using cohen's D/Hedges'g $> = 0.5$)

[1]compared to respondents who eat meat once a week

[2]compared to male respondent

[3]compared to those respondent with Master/PhD degree

[4]compared to those who live with their parents.

orientation correlates to a lower overall acceptability for harming animals (b = -0.18, p<0.01 in model 1; and, b = -0.11, p<0.05 in model 2). This means that when holding all other variables constant, one point increase in IP is more likely to decrease 0.18 point of AIS score in the first, and 0.14 point in the second model. However there is one difference between both models whereby the effect-size of IP shows small effect-size toward AIS score in the first model ($0.02 < = F^2 < 0.15$), but rather small to no effect in the second model ($F^2 < 0.02$). Through the confidence interval, if we were to retake both models for total of 20 random trials, taking samples of the same size from the same population, we can be confident that for 19 out of total 20 trials (95% of the time), an increase of one unit in IP will be more likely to decrease AIS between -0.25 to -0.12 point in the first model, while in the second model will be more likely to decrease AIS between -0.20 to -0.014 point.

### 3.6. Extrinsic social religious orientation, ethical ideologies, and acceptability for harming animals

The hypothesis presented in this section is that a higher extrinsic social religious orientation (ES) correlates to lower idealism (I), higher relativism (R), and a higher acceptability for harming animals (AIS). We find only partial support to the fourth hypothesis. Table 6 shows that higher extrinsic social religious orientation correlates to a higher relativism (r[927] = 0.15, p<0.01), but not to a lower idealism (r[927] = -0.02, p>0.05). In Table 7, using multiple regression, we confirm that higher extrinsic social religious orientation relates to a higher overall acceptability for harming animals in both the first (b = 0.12, p<0.01) and the second model (b = 0.05, p<0.05). This means that when holding all other variables constant, one point increase in ES is more likely to increase 0.12 point of AIS score in the first, but only 0.05 point in the second model. However there is one difference between both models whereby the effect-size of ES shows small effect-size toward AIS score in the first model ($0.02 < = F^2 < 0.15$), but rather small to no effect in the second model ($F^2 < 0.02$). For the confidence interval, if we were to re-fit both models for total of 20 random trials, taking samples of the same size from the same population, we can be confident that for 19 out of total 20 trials (95% of the

time), an increase of one unit of ES will be more likely to increase AIS between 0.08 to 0.15 point in the first model, while in the second model will be more likely to increase AIS between 0.007 to 0.095 point. Therefore, except for every relations with idealism, the present study accepts all the expected main variables' relations in the hypothesis.

### 3.7. Demographic and other determinants

In the second regression model (*see* Table 7), aside the main variables, there are some demographic and other determinants closely related to AIS, which are meat consumption (b = 0.36, p<0.01), gender (b = -0.14, p<0.01), diploma (b = 0.39, p<0.05) education level and living in own house home ownership (b = 0.11, p<0.05).

## 4. Discussion

Three general conclusions are supported by the present study: *first*, two components of religious orientation relate to ethical ideologies. Intrinsic personal religious correlates with idealism, and extrinsic social religious orientation correlates with relativism. This evidence leans more towards the study by Watson et al. [24], stressing the relationship between religious orientation and ethical ideologies, rather than the study by Barnett et al. [21], stressing religiosity related only to ethical relativism. However, in another vein, the present study differs greatly from Watson et al. [24], who stated that "..intrinsicness seemed to reflect an idealistic and anti-relativistic religious identity" (p. 160). In contrast, with intrinsic and extrinsic social religious orientation, this study provide evidence for the connection of religiousness to idealism and relativism. *Second*, rather than idealism, observing the first regression model, we find that only ethical relativism relates to the acceptability for harming animals in the predicted direction, which strengthens the role of relativism found in previous studies [16–20]. However, from the second model there are no significant relation between ethical ideologies and AIS, and therefore we stress religious orientation as a more consistent predictor to the acceptability for harming animals. *Third*, both the intrinsic personal and extrinsic social religious orientation, as hypothesized, consistently relate to the acceptability for harming animals. However, contrary to previous studies, we find no support for the relation between the treatment of animals with religious inspiration [20], and with religious affiliation [20, 28, 29].

Lastly, by including common important determinants—consistently suggested by previous studies—in the regression of the main variables, this study presents a critical evaluation for the correlation of all the main variables' relations. Each set of the result for ROS and EPQ towards AIS are discussed in the respective sections.

### 4.1. Ethical ideologies to AIS

Taking only the main variable as predictor in the first model, except for idealism, this study confirms Su & Martens [18] findings whereby higher relativism significantly correlated with higher acceptability for harming animals [18, 83, 84]. For ethical idealism, this study produces mixed results, which are not always in agreement with Su & Martens [18]. As we reported previously, to overall AIS total score, there is no significant relation from ethical idealism. However, observing regression results of only the main variables (model 1) to each of AIS' sub-issues (*see* Tables 1–7; in S1 File), on the one hand, for 'killing animals and animal welfare deprivation', 'harming animal for environmental' and 'harming animal for social' issues, the result suggests that the more the respondents consider their ethical behaviour will lead to desirable consequences (a high score of ethical idealism), the lower their acceptability toward harming animals. On the other hand, the reverse happens in 'animal use', 'experimentation on animals', and 'animal genotype change'. While Su & Martens [18] proposed that the association is most

likely due to the idealist's reluctance to overlook animal suffering [25] which relates to empathy, this study suggests that it also closely depends on the core motives for harming animals. Provided with the motives and reasons for harming animals, it seems that people may view differently of what is considered as ethical and non-ethical behaviour. Nevertheless, from the first regression model (Tables 1–7; in S1 File), the significance of idealism towards various AIS subscales is rather ambiguous.

For ethical relativism, the present study finds relationship between relativism and AIS and therefore replicates and strengthens Su & Martens [18] findings where they reveals that a high level of ethical relativism more likely to lead to a higher acceptability for harming animals. In the first model, compared to idealism, relativism acts as stronger predictor for acceptability for harming animal. The more the respondents view multiple ways and principles undergirding their judgement and decision-making, they are more likely to accept harming animals. This result is also consistently true in most of AIS' subscales, namely, 'integrity destruction', 'killing animals and animal welfare deprivation', 'harming animal for environmental' and 'harming animal for social' issues. Only in 'animal use', 'animal experimentation' and 'animal genetic change' issues, this study finds no significant role of relativism.

Nevertheless, as one important addition, through the second model, this study offers a new insight of the non-existent ethical ideologies relation to the acceptability for harming animals when including other competing factors. With the account of demographic and other determinants, this study shows that compared to religious orientation, ethical ideologies are simply have no role in predicting acceptability for harming animals.

## 4.2. Religious orientation to AIS

White's [32] study marked a milestone where research of religions' relationship with environmental sustainability began. In that growing research field, related to the aspect of belief [85], end-times theology [35], or belief in either an afterlife or divine intervention [86], a broad swathe of evidence has shown that religion depresses concerns for the environment [33, 34] and religious believers' were found to have a relatively low perception of urgency for environmental issues. Examining religious orientations' relationship to the acceptability for harming animals, the present study do not find unanimous evidence supporting White's [32] thesis. Respondents with high IP are more likely to have a lower acceptability for harming animals. Rather than hindering the importance of animal protection, religious belief and the degree to which religion is internalized into respondents' everyday conduct has been found to enhance respondents' perceptions of the importance of animal protection. By way of explaining this mixed result, the present study suggests that individuals' interpretation of religious scripture as the result of communication framing may be important [43, 87]. One study has pointed out that reframing environmental discourse in multiple religious teaching interpretations reduces the gap in environmental concern between liberals and conservatives [87]. In another study, religious framing of climate change resonates with the electorates of both progressive and conservative politicians and serves as a bridging device for bipartisan climate-policy initiatives [43]. Hence, this study suggests that providing information about, or controlling for, multiple religious teaching scenarios is important to further explaining variation between different research results.

On the other hand, the ES religious orientation dimension supports White's [32] thesis whereby religion depresses concerns about ecology and also, therefore, about animals. Individuals who have high ES showed a higher acceptability for harming animals. The construct of ES implies religion serves as an instrument for social gain, exemplified by the membership of a powerful in-group, providing protection, consolation and social status, allowing religious participation, or use of an ego defence [52, 54–57]. Thus, ES properties appear to more closely

resemble the embodiment of social identity theory, rather than that of religious belief and commitment. Therefore, the present study may actually reveal how the social identity aspects of religion (for example, religious group affiliation, participation, and the like) [21] can hinder concern for the environment.

Lastly through the second model, the present study stresses the consistent relationship between religious orientation with acceptability for harming animals. Even when taking into account all other variables including demographic and other important determinants, religious orientation remains consistent in predicting acceptability for harming animals.

### 4.3. ROS an EPQ to AIS

Other than unearthing important evidence for ethical relativism, perhaps one of the more significant contributions from the present study is that it examines also the main correlation of religious orientation components (IP and ES) and ethical ideology components (idealism and relativism), all taken together, as well as independently.

Contrary to prediction, IP does not have a significant relationship with relativism. This is surprising considering that the sample mean indicated that most of the respondents considered themselves to be very strongly committed to their religious beliefs (IP Mean of 4.22 with maximum score of five) suggesting that having a strong, deep religious belief and commitment does not necessarily mean that respondents consider them as their sole governing universal moral guiding principle for their judgement and decision-making. Furthermore, IP correlates with idealism [88]. This may suggest that rather than operating as the extent to which an individual believes in universal governing moral principles (low relativism), intrinsic personal religious motives, belief and commitment may function more as a principle with which individuals portray and justify their actions as correct, in order to achieve desirable outcomes (high idealism).

*Second*, ES relates to relativism. The more individuals view their religious belief, participation and practices as the means to an end for social motives and affiliation (for example, as group protection, group status, or other means of social gain), the more likely they are to have high relativism. High relativistic individuals' moral judgments are adaptable, for they base their appraisals on features of the particular situation and action they are evaluating. People who express low relativism, in contrast, have more cognitive beliefs in universal moral principles, and use them to make judgements and decisions [87] (p. 815).

It is interesting to note that an unexpected positive correlation was observed between idealism and relativism (r[927] = 0.35, $p<0.01$). This is contrary to the original EPQ study which suggested that the two scales were essentially orthogonal [21, 27]. Moreover, this unexpected correlation was also shown in Barnett, Bass, and Brown [21] when investigating the relation between EPQ and religiousness. Their study suggested consistent evidence of the psychometric limitations of ethical idealism and relativism constructs when presented and measured on a single scale [88].

Lastly, when taking into account of all the main variables with demographic and other important determinants, the results stress the importance of religious orientation as the sole main variable that relates to acceptability for harming animals. Both idealism and relativism do not have any correlation to acceptability for harming animals in this model. This finding strongly suggests religious orientation as the more prominent main variable in predicting acceptability for harming animals.

### 4.4. Demographics and other determinants

Demographic factors like meat consumption, gender, level of schooling, and type of home ownership are significant with respect the overall AIS score (*see* Table 7). However by

examining the effect size, only meat consumption, gender, and diploma level of schooling are discussed.

Gender was often found to be a correlated factor [77, 89, 90] and the present study replicated those findings. After all demographic and other factors are taken into account, this study reveals gender as one of persistent predictor for AIS. In one study, women are regarded as being more concerned with animal welfare than men [25]. On the other hand, personality differences between gender may play important roles as one study suggest that the differences whereby men are less likely to have sympathetic reactions to animals than women are probably derived from men's lower levels of belief in the mental abilities of animals compared to women [91].

The next important demographic determinant is how often respondents consume meat in a given week. The result shows that, compared to those who consume meat once per week, individuals who do not consume meat have higher acceptability for harming animals. It is difficult to explain this result without fully understand the respondents' monthly income. Unless this result originates from being conscious of leading a healthy life, or from the motive to preserve the natural environment, answering no meat consumption in their daily diets voices a very different meaning when it is in the context of low monthly income category. However, related to monthly income and expenses, the present study finds no significant relation in the regression model. Cross-checking with ANOVA, this study finds significant difference between income categorical groups (F[5] = 2.50, $p$ = 0.029). However, the post-hoc tests using Bonferroni method shows no significant difference between income group categories. One possible cause may rest in how this study allows participants to choose 'refuse to answer' option to answer the monthly income question. It is possible that respondents from both highest and lowest monthly income may refuse to answer this specific question, and thus, blurs whatever group difference that may be found otherwise. Therefore, this study does not yet have a sufficient explanation other than to carefully propose that meat consumption may warrant further investigation by examining how it may relate to monthly income.

The present study also indicates that the level of schooling correlates with the overall acceptability for harming animals. Specific to this, result shows that compared to respondents with a Master/PhD degree, those respondents who have a diploma as their last level of schooling have higher acceptability for harming animals. One probable explanation is that participants with higher level and more advance degree like Master or PhD may have more exposure and access to environmental and animal welfare information, compared to diploma degree which usually revolves more around pragmatic and technical skills.

## 4.5. Limitations

Despite the present study's success in examining EPQ and ROS along with influential factors for the acceptability for harming animals, it is clear that meat consumption and home ownership variables remain unexplained. For the latter, findings show that respondents who live in their own house are more likely to have a higher acceptability for harming animals compared to those who still live with their parent. It may be possible that having own house refers to an older, more mature and more pragmatic respondents having more responsibilities for their livelihood compared to younger respondents who still live with their parents. However considering the small effect-size, the present study suggests the need for a deeper effort in deploying follow-up interviews to gain insight into how those variables may or may not necessary relate to the primary variables. Lastly, the present study only finds partial evidence that acceptability for harming animals correlates positively with ethical relativism, as it was reported by Su & Martens [18]. However, the remaining parts unearthed with this study is the consistent roles

of religious orientation, even more significant than ethical ideologies. Previous studies confirm that the mechanisms underlying the relation of ethical idealism and relativism to attitudes toward animals may vary in different countries and cultures [88]. Nevertheless, the present study provides further insight and introduces religious orientation as one of the contributing cultural factor that warrants further investigation.

## 4.6. Animal welfare implications

The present study highlights the significant relationship between religious orientation and relativism to AIS. Regarding relativism, the results imply that individuals who believe in a universal governing moral principle are more likely to have a higher awareness of animal protection, and, therefore, a lower acceptability toward harming animals. For religious orientation, results imply that individuals who have deep personal religious belief and commitment to their religion would likely have a low acceptability for harming animals. However, when people have extra ulterior motives of for pursuing social gain, status, affiliation, or membership with their religious activities participation, it would be more likely that they have a higher acceptability for harming animals. Thus, the present study not only supports previous findings [18, 20], but also contributes to addressing religious orientation as a significant variable closely related to attitudes towards animals. Perhaps, one additional contribution of this study is that it may help to explain some mixed results in studies investigating White's (1967) thesis about the inhibiting influence of religion to environment preservation action and effort.

In addition, the present study extends the potential for animal protection awareness to reach broader platforms, for example, in the case where religious values and institutions could serve as motivational platforms. One key implication of these results is the need to examine how religious orientation interacts with ethical ideology in affecting people's positive attitudes towards animals. Finally, as this is the first paper to investigate how both religious orientation and ethical ideology relates to animal protection, other research focusing on specific animals such as companion animals [9, 92], carnivores [93], or animals important to maintaining ecosystem health for environmental sustainability, may be introduced as focal points in religious studies and related platforms.

## Supporting information

**S1 Appendix. Questionnaires in English.**
(DOCX)

**S2 Appendix. Questionnaires Bahasa Indonesia adaptation.**
(DOCX)

**S1 Data. Missing case analysis, factor analysis and reliability.**
(DOCX)

**S1 File. AIS subscales multiple regression results.**
(DOCX)

## Acknowledgments

We acknowledge the significance of Indonesia Endowment Fund (Lembaga Pengelola Dana Pendidikan Indonesia), the Faculty of Psychology Universitas Indonesia, Rakata Alam Terbuka Foundation, Universitas Islam Malang, Faculty of Psychology Universitas Brawijaya, the AnimalWise Foundation, Yeka Kusumajaya, and KH. Ahmad Zubaidah (Gus Ida), for their enduring support in various aspects of this research. We thank all the respondents for their

participation in this survey. Also, the work by P. Martens has partly been made possible by the fellowship 'Ethics of the Anthropocene', Free University Amsterdam.

## Author Contributions

**Conceptualization:** Dexon Pasaribu, Pim Martens, Bagus Takwin.

**Data curation:** Dexon Pasaribu.

**Formal analysis:** Dexon Pasaribu, Pim Martens, Bagus Takwin.

**Funding acquisition:** Dexon Pasaribu.

**Investigation:** Dexon Pasaribu.

**Methodology:** Dexon Pasaribu, Pim Martens, Bagus Takwin.

**Project administration:** Dexon Pasaribu.

**Resources:** Dexon Pasaribu.

**Supervision:** Pim Martens, Bagus Takwin.

**Validation:** Dexon Pasaribu, Pim Martens, Bagus Takwin.

**Writing – original draft:** Dexon Pasaribu.

**Writing – review & editing:** Dexon Pasaribu, Pim Martens, Bagus Takwin.

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
