## [Decision Letter · Decision Letter 0]

24 Jul 2020

PONE-D-20-16700

Do religious beliefs influence concerns for animal welfare? The role of religious orientation and ethical ideologies in attitudes toward animal protection amongst Muslim teachers and school staff in East Java, Indonesia

PLOS ONE

Dear Dr. Pasaribu,

Thank you for submitting your manuscript to PLOS ONE. After careful consideration, we feel that it has merit but does not fully meet PLOS ONE’s publication criteria as it currently stands. Therefore, we invite you to submit a revised version of the manuscript that addresses the points raised during the review process.

I would normally solicit at least two reviews, but I agree very much with the reviewer, and have decided to expedite the decision-making process to save us all some time. 

We look forward to receiving your revised manuscript.

Kind regards,

Jonathan Jong, PhD

Academic Editor

PLOS ONE

Journal Requirements:

3. Please ensure that you include a title page within your main document. We do appreciate that you have a title page document uploaded as a separate file, however, as per our author guidelines (http://journals.plos.org/plosone/s/submission-guidelines#loc-title-page) we do require this to be part of the manuscript file itself and not uploaded separately.

Could you therefore please include the title page into the beginning of your manuscript file itself, listing all authors and affiliations

4.We note that you have indicated that data from this study are available upon request. PLOS only allows data to be available upon request if there are legal or ethical restrictions on sharing data publicly. For more information on unacceptable data access restrictions, please see http://journals.plos.org/plosone/s/data-availability#loc-unacceptable-data-access-restrictions.

5. Please upload a new copy of Figure 1 and 2 as the detail is not clear. Please follow the link for more information: https://blogs.plos.org/plos/2019/06/looking-good-tips-for-creating-your-plos-figures-graphics/" https://blogs.plos.org/plos/2019/06/looking-good-tips-for-creating-your-plos-figures-graphics/

6. Please include captions for your Supporting Information files at the end of your manuscript (Appendix 1 and 2 file type changed from "other" to "supporting information"), and update any in-text citations to match accordingly. Please see our Supporting Information guidelines for more information: http://journals.plos.org/plosone/s/supporting-information.

Reviewers' comments:

Reviewer's Responses to Questions

**Comments to the Author**

1. Is the manuscript technically sound, and do the data support the conclusions?

Reviewer #1: Partly

2. Has the statistical analysis been performed appropriately and rigorously? 

Reviewer #1: No

3. Have the authors made all data underlying the findings in their manuscript fully available?

Reviewer #1: No

4. Is the manuscript presented in an intelligible fashion and written in standard English?

Reviewer #1: Yes

5. Review Comments to the Author

Reviewer #1: This paper presents the results of a really nice dataset. In a sample of 929 teachers in East Java, Indonesia - this study probes the relationships between religiosity (intrinsic/extrinsic), broad concern for animal welfare, and acceptability of animal use across a number of dimensions. While I commend the authors for their efforts in generating such a large dataset with the potential to make an informative contribution to the literature, I think this paper requires lots of additional work before it will be suitable for publication - and especially so (but not exclusively) with regards to providing more detail into how the data was handled. That being said, it’s important to reiterate that this is likely very valuable data - the authors will just have to do more to convince readers that they have properly tested their focal hypotheses. My primary concern is that given the manuscript in its current state, I would not be able to (1) replicate the study should I be interested in doing so as their methods are underspecified and (2) even if I had access to their already collected data, I’m not convinced that I would be able to reproduce the analyses from the details presented in this paper.

¬

The authors indicate that 9-items from the 20-item animal attitude scale (AAS) were dropped because it increased internal consistency. However, the authors never provide details as to which items were dropped - nor do they inform readers as to exactly how reliability was improved by dropping these items. Moreover, might it be the case that the full set of items are simply better reduced to multiple factors as opposed to one? The results of an EFA on all items (either in the main text or a supplemental materials section) would go a long way to demonstrating the decisions made in creating one of the focal outcome variables in this study. The same is true for the Ethical Position Questionnaire, where 5 items were dropped from the full scale.

In discussing their sample, the authors say that 78 respondents were dropped because of missing data; but that their data was replaced using “linear trends method”. Their final sample size is 929 (=1007 minus 78) and thus it doesn’t seem like any data was replaced? The authors should be more clear about the imputation process they used, if they used one. Minor comment, including an age range would be helpful.

The authors state that some of their data was “translated into a normal distribution using either outlier removal, log-10 or square-root transformation” (line 239). But yet, they never tell readers which variables have been transformed. More detailed information about the spread of responses (histograms or density plots, for example) would go a long way in justifying these decisions to transform the data. But more importantly, not knowing which variables were transformed makes the focal tests of the hypotheses presented in the regression impossible to interpret - as readers have no insight into how the variables were entered into the model.

On that front, are the regression coefficients presented in Table 3 standardized? If not, were the variables centered (which would be necessary to make sense of any of the coeffecients given the interactions included in the models). Moreover, the authors focus exclusively on statistical significance and don’t interpret their effect sizes - most of which seem really small (although again, impossible to interpret given the lack of information about how the variables were coded and entered into the model). I’d suggest that confidence intervals should be included around all effect size estimates in the manuscript (including correlation tables, regression tables, etc.) - and more should be done to interpret the magnitude of associations rather than just their statistical significance. In a large enough sample such of this one, most non-zero effects will be significant - but it is up to the authors to convince readers that these are in any way meaningful associations.

As the analysis plan was not pre-registered, I expect to see more than one regression model per outcome variable. I’d suggest one regression table per outcome measure - with several models included for each. For instance, an informative table might present the results as follows: Model 1 (main predictor variables), Model 2 (main predictors + demographic controls), Model 3 (main predictors + demographic controls + interaction variables). This way, any variation in effect size estimates related to inclusion/exclusion of other covariates can be made explicitly clear to readers; and provides a good test of how robust effects are to addition of controls. One way to limit the number of tables here would be to move sub-scale analyses on the Animal Issues Scale to a supplemental materials section. Given the high reliability of scores across the sub-scales (.93) - I’m not convinced that authors are gaining all that much from repeatedly testing their hypothesis at the sub-scale level. Moreover, this would really help streamline the results section which is, at the moment, quite dense.

In the regressions, the authors test for the 3-way interaction between forms of religiosity, and the interaction between idealism/relativism. However, neither of these interactions seem to follow directly from the predictions laid out in the introductory sections. Thus, I think it would be important to show models with and without these interactions included. Or, if there are no strong predictions to be made about these interactions - simply omit them for now (or move tests of them to a supplemental materials section).

In following up on these interactions, the authors present an ANOVA assessing whether outcome measures differ based on level of idealism/relativism. However, the results of the regressions where this interaction was consistently not significant would indicate that there is no interaction between these variables across outcomes - and thus retesting this interaction in the ANOVA is, I think, unwarranted. I would advise against breaking up the data in this way. This is especially true given that the authors don’t provide details as to how their absolutist/situationist/exceptionist/subjectivist categories were created - and thus I think these additional analyses do little to elaborate on what is already being tested in the regression. Lastly, in plotting these interactions - the full scale of response options should be presented on the y-axis, as the zoomed in scales presented here make a very tiny effect seem bigger than it is. Moreover, it would be helpful if raw data points were also plotted and not just the lines.

As mentioned before, the results section is already quite dense. One way I see to really alleviate some pressure on readers is to move all the writing about demographic covariates to a supplemental materials section. Although surely valuable, these are not the focal predictions of the paper and including them in the main results sections both detracts from the tests of the focal hypotheses about religion and makes the results section harder to digest. This would helpfully reduce the length of the discussion section as well.

I think the introduction would benefit greatly from some further specification of the reviewed literature. Indeed, the authors mention that much of the discrepancy or mixed results might result from different operationalizations of religion/religiosity in different samples. But yet, they do little to highlight differences in methodologies in reviewing the literature - and importantly, it is rarely the case that the “religion” being investigated in the literature is specifically mentioned - making it very hard to track what has already been done in this research area. It’s important to not generalize across religious traditions, and would be very helpful for readers if the authors were more specific in their review. I think the clarity in the section discussing ethical idealism v ethical relativism could serve as a good model for how the rest of the introduction’s review should be structured. Lastly, I would caution quite strongly against the use of “proven” in describing the results of any empirical research, and there are several instances when causal language is used to describe the results of correlational research.

All in all, behind this paper there is good and useful data and with some more work it will be a really nice paper. I was surprised that there was not more discussion of the fact that this data was from teachers in the discussion, and would be curious to see more reflection on that from the authors in a revision.

6. PLOS authors have the option to publish the peer review history of their article (what does this mean?). If published, this will include your full peer review and any attached files.

Reviewer #1: No

---

## [Author Response · Author response to Decision Letter 0]

10 Dec 2020

I have wrote all my response to each of the issue you're addressing in the attached file: 'response to reviewer.'

Thank you for all your critiques, help, feedbacks and inputs, dear reviewers.

---

## [Decision Letter · Decision Letter 1]

23 Mar 2021

PONE-D-20-16700R1

Do religious beliefs influence concerns for animal welfare? The role of religious orientation and ethical ideologies in attitudes toward animal protection amongst Muslim teachers and school staff in East Java, Indonesia

PLOS ONE

Dear Dr. Pasaribu,

Thank you for submitting your manuscript to PLOS ONE. After careful consideration, we feel that it has merit but does not fully meet PLOS ONE’s publication criteria as it currently stands. Therefore, we invite you to submit a revised version of the manuscript that addresses the points raised during the review process.

We look forward to receiving your revised manuscript.

Kind regards,

Anima Nanda

Academic Editor

PLOS ONE

Reviewers' comments:

Reviewer's Responses to Questions

**Comments to the Author**

1. If the authors have adequately addressed your comments raised in a previous round of review and you feel that this manuscript is now acceptable for publication, you may indicate that here to bypass the “Comments to the Author” section, enter your conflict of interest statement in the “Confidential to Editor” section, and submit your "Accept" recommendation.

Reviewer #1: (No Response)

2. Is the manuscript technically sound, and do the data support the conclusions?

Reviewer #1: Yes

3. Has the statistical analysis been performed appropriately and rigorously? 

Reviewer #1: Yes

4. Have the authors made all data underlying the findings in their manuscript fully available?

Reviewer #1: Yes

5. Is the manuscript presented in an intelligible fashion and written in standard English?

Reviewer #1: Yes

6. Review Comments to the Author

Reviewer #1: (No Response)

7. PLOS authors have the option to publish the peer review history of their article (what does this mean?). If published, this will include your full peer review and any attached files.

Reviewer #1: No

---

## [Author Response · Author response to Decision Letter 1]

14 Apr 2021

We wrote our responses for all the reviewers' questions in a specific file: '2ndroundreview_responsetoreviewer.docx', and uploaded it along with the manuscript revisions.

---

## [Editor Report · Decision Letter 2]

7 Jul 2021

Do religious beliefs influence concerns for animal welfare? The role of religious orientation and ethical ideologies in attitudes toward animal protection amongst Muslim teachers and school staff in East Java, Indonesia

PONE-D-20-16700R2

Dear Dr. Pasaribu,

We’re pleased to inform you that your manuscript has been judged scientifically suitable for publication and will be formally accepted for publication once it meets all outstanding technical requirements.

Kind regards,

Anima Nanda

Academic Editor

PLOS ONE
---

## [Editor Report · Acceptance letter]

9 Jul 2021

PONE-D-20-16700R2 

Do religious beliefs influence concerns for animal welfare? The role of religious orientation and ethical ideologies in attitudes toward animal protection amongst Muslim teachers and school staff in East Java, Indonesia 

Dear Dr. Pasaribu:

I'm pleased to inform you that your manuscript has been deemed suitable for publication in PLOS ONE. Congratulations! Your manuscript is now with our production department. 

Kind regards, 

on behalf of

Dr. Anima Nanda 

Academic Editor

PLOS ONE